# Changes in Attitudes toward People with Substance Use Disorder: A Comparative Study of the General Population in Mexico

**DOI:** 10.3390/ijerph19148538

**Published:** 2022-07-13

**Authors:** Marycarmen Bustos-Gamiño, Jazmín Mora-Ríos, Jorge Villatoro-Velázquez, Clara Fleiz-Bautista, Alejandro Molina-López, María Elena Medina-Mora

**Affiliations:** 1Dirección de Investigaciones Epidemiológicas y Psicosociales, Instituto Nacional de Psiquiatría Ramón de la Fuente Muñiz (INPRFM), Mexico City 14370, Mexico; marybustos@imp.edu.mx (M.B.-G.); ameth@imp.edu.mx (J.V.-V.); fleiz@imp.edu.mx (C.F.-B.); metmmora@gmail.com (M.E.M.-M.); 2Seminario de Estudios de la Globalidad, Facultad de Medicina, Universidad Nacional Autónoma de México (UNAM), Mexico City 04510, México; 3Sudirección de Servicios Clínicos, Instituto Nacional de Psiquiatría Ramón de la Fuente Muñiz (INPRFM), Mexico City 14370, Mexico; doctor.alex.psiquiatra@gmail.com; 4Director of Faculty of Psychology, Universidad Nacional Autónoma de México (UNAM), Mexico City 04510, Mexico

**Keywords:** attitudes toward people with addictions, epidemiological surveys, public stigma, Mexico

## Abstract

Background: Substance use disorders are among the most stigmatized conditions worldwide. People with substance use disorder (PWSUD) are often considered responsible for their use of drugs. The objectives are to analyze changes in Mexican attitudes toward PWSUD in the general population over the period 2011 to 2016 and to use the latest Mexican household survey to determine which segments of the population are most likely to have negative attitudes. Methods: Two representative national household surveys employing similar methodologies were conducted in Mexico in 2011 and 2016 with persons aged 12–65 years. Participants were asked about their attitudes toward PWSUD, and changes were compared across GLM. Results: The surveys found a decrease from 2011 to 2016 in the number of respondents who considered PWSUD “sick” or in “need of help” and an increase in the number who believed they were “selfish” or “criminal”. The 2016 survey found that men, people 18 years of age or older, people who do not use drugs and people with lower educational levels were the groups with the most negative attitudes toward PWSUD. Conclusions: These results suggest that it may not be recognized that PWSUD may have a health problem and that this helps to increase stigmatization towards this population.

## 1. Introduction

Researchers have characterized stigma as a social or public phenomenon. People with a particular attribute are the object of rejection using negative descriptions and judgments that affect their quality of life and well-being. One of society’s most stigmatized health conditions is substance use disorder (SUD) [1].

### 1.1. Stigma toward PWSUD in the World

A World Health Organization study with 14 countries found that SUD was among the most highly stigmatized of 18 conditions [2]. The stigma associated with SUD affects different population groups and is expressed at various social levels, including in the family, the community and health care institutions, and it is a problem in various parts of the world [3,4,5,6,7,8,9,10,11].

In Europe and Latin America, several studies have indicated that persons with substance use disorder (PWSUD) are highly stigmatized by the public and experience a higher stigma than those with psychiatric disorders. The PWSUD are considered “criminals”, weak in character, alienated, lacking self-control and irresponsible, which generates emotional reactions of fear, anger, pity and rejection from the general public, family and health professionals [1,3,4,5,9,10,12]. These responses can lead PWSUD to internalize beliefs and feelings of unworthiness that affect their self-image, functioning, self-sufficiency and mental health, blocking access to early treatment for those interested in reducing their substance use [13,14,15,16,17].

Similarly, the Spanish Foundation for Assistance Against Drug Addiction analyzed the perceptions of PWSUD over a period of ten years (2004–2014) and found that most of the population leaned toward negative attitudes, especially women, those aged 50 or over, members of religious groups and those with lower educational levels. On the other hand, those with more favorable attitudes were men, those on the ideological left and young people, who were more inclined to support the adoption of less restrictive laws concerning alcohol and cannabis. Additionally, a third group was identified that supported policies that emphasized the treatment of such problems [18].

### 1.2. Studies in Mexico

In Mexico, some related studies examined the social perceptions of different types of mental illness and found that the adult population most rejected those with schizophrenia and SUD [19].

Another study found evidence of criticism, mockery, inappropriate comments, overprotection, hostile attitudes, and verbal and non-verbal aggression aimed at PWSUD from health care personnel working in the area of drug use disorder, as well as with PWSUD and their families.

Although these were present among different kinds of social groups, they were found mainly among unregulated self-help groups. The general population tends to have negative attitudes toward PWSUD, which leads to the perception of drugs as a moral rather than a public health issue and attributing responsibility, as well as criminality and weakness of character, to PWSUD [20].

The 2016 National Household Survey on Drug, Alcohol and Tobacco Use in Mexico found that 0.6% of the total population had drug dependency, but that 21% of those persons had not sought treatment for fear of being identified, and 14.3% had not sought it because they were concerned about what others would think if they knew they were in drug treatment [21].

### 1.3. Aim of This Study

The international trend in recent years has been toward facing the consequences of punitive policies based on prohibition and control [22]. Given this, in Mexico, there are few studies on attitudes toward PWSUD, particularly in national probabilistic studies in the general population.

The relevance of this study (which analyzes two probabilistic studies on a national level for the first time) is to have national results of the problem that aid in developing psychoeducational interventions with the general population that reduce stigma and discrimination toward PWSUD.

Therefore, the first objective of the present study is to analyze changes in Mexican attitudes toward PWSUD related to several sociodemographic variables in the general population over the period 2011 to 2016.

Additionally, the second objective is to analyze the latest Mexican household survey regarding the use of drugs, carried out in 2016, to determine which segments of the population are most likely to have negative attitudes toward PWSUD.

## 2. Materials and Methods

The data analyzed are from the 2011 National Household Survey of Addictions (Encuesta Nacional de Adicciones, [ENA] 2011) [23] and the 2016 National Household Survey of Drug, Alcohol, and Tobacco Use (Encuesta Nacional de Consumo de Drogas, Alcohol y Tabaco (ENCODAT), 2016) [21], carried out in the homes of respondents aged 12–65 years in rural and urban communities in Mexico. The surveys used similar methodologies, which allows for comparison [21,23].

These household surveys evaluate the prevalence, consumption and problems associated with the consumption of illegal, legal and medical drugs for the purpose of formulating and implementing public policy in Mexico, a large, culturally diverse country with a high degree of socioeconomic inequality. Mexico has a population of 126,014,024 people (48.8% male and 51.2% female), of whom 49.3% have an elementary school education. Its Human Development Index is 0.767, ranking 76th out of 189 countries, and the national birth rate is 2.1% [24]. Culturally, there is greater acceptance in Mexico for alcohol than for drug consumption [25,26].

### 2.1. Design and Participants

The surveys were cross-sectional, with a multi-stage, probabilistic, and stratified design and a confidence level of 90%. The sampling universe for the primary sampling units (PSUs) was the sum of the Basic Geographical Statistical Areas (BGSAs), stratified according to state and urban-rural character. Participants were 12–65 years of age, from urban and rural communities, and living at home. Wherever possible, following the household questionnaire, one adult aged 18–65 and one teenager aged 12–17 were presented with the individual questionnaire, according to a simple random sampling in each age group. The ENA 2011 was nationally representative. The total response rate (household + individual questionnaires) was 73.3%, with a final sample of 16,249 complete interviews: 7859 men and 8390 women; 2742 teenagers and 13,507 adults. The ENCODAT 2016 was nationally representative, with a total response rate (household + individual) of 73.6% and a final sample of 56,877 complete interviews: 27,463 men and 29,414 women; 9563 teenagers and 47,314 adults. Combining the two yielded a sample of 73,126 individuals: 35,322 men and 37,804 women; 12,305 teenagers and 60,821 adults.

Both surveys were evaluated and approved by the Research and Ethics Committee of the Instituto Nacional de Psiquiatría Ramón de la Fuente Muñiz (Approval Nos. CEI/12/04/11 and CEI/083/2015). Participants were read a statement of informed consent, and information was collected only from those who agreed to participate. Consent for minors was requested from their parents or guardians, who signed written consent forms; the minors themselves also agreed to participate. Respondents were assured of the voluntary nature of their participation and the confidentiality of the information.

Additionally, interviewers, supervisors, cartographers, computer support personnel and coordinators were trained for each of the surveys. The surveys were carried out simultaneously in different states. Interviewers presented the household questionnaires in selected homes with the help of a computer and then presented the individual questionnaire to the adults and teenagers selected at random by the computer application. The supervisors verified with each of the households recorded as declining to participate in the study that this was, in fact, the case. Each home selected was visited at least four times on different days and at different times, including on weekends, to increase the probability of finding household members at home.

### 2.2. Questionnaire

Information was obtained through a questionnaire presented to a randomly selected individual. Sociodemographic data were requested, as well as information about tobacco and alcohol use, use of illegal drugs and prescription drugs for non-medical purposes, problems related to substance use and mental health of the respondent. Both surveys, like previous surveys, were conducted in face-to-face interviews using a computer.

The following variables were considered in both surveys.

Age: Respondents were grouped into three categories: 12–17 years (teenagers) (code 1), 18–29 years (young adults) (code 2) and 30–65 years (adults) (code 3).

Persons with drug use lifetime: These were defined as respondents who reported having ever used illegal drugs (code 1), such as marijuana, cocaine, crack, hallucinogenics, inhalants, heroin, methamphetamines, ketamine or gamma-hydroxybutyrate (GHB); or medical drugs for non-prescription purposes, such as opiates, tranquilizers, sleeping pills, barbiturates or amphetamines. Those who had never used any of these drugs were classified as persons without drug use, or non-users, with code 0.

Community: Rural communities were defined as those with fewer than 2500 inhabitants (code 1); those with 2500 or more were considered urban (code 2).

Attitudes toward PWSUD: This indicator was evaluated in the 2011 survey with questions beginning with “For you, is an addict a person who is…?”, while in 2016, the question was slightly different: “In your opinion, is a person with addiction to drugs…?” These questions were then posed for five different attributes to elicit yes (code 1) and no (code 0) responses. (In these surveys (2011 and 2016), the word “addict” or “person with addiction” to drugs was used because it was easier to understand those terms for the general population. Additionally, at that time, at least for the 2011 survey, international classifications did not use the term person with substance use disorder.) These attributes included “weak,” “selfish” and “criminal” to represent the negative attitudes associated with a moral and punitive vision of drug dependence, and “in need of help” and “sick” to represent the sympathetic attitudes that view it as a treatable condition requiring specialized treatment. It is important to note that these categories of analysis are from prior studies of attitudes toward PWSUD, as evaluated in the national survey [27], with a Cronbach’s alpha of 0.70 and factor loadings for each item greater than 0.40.

### 2.3. Statistical Analysis

Statistical analyses were carried out using the program STATA, version 13. Estimations were obtained of the prevalence of each of the attitudes toward PWSUD. To assess the changes between surveys, an analysis of prevalence ratios (PR) was carried out based on generalized linear models (GLM) with log-link and binomial distribution [28]. From this global analysis, the nlcom command was used to obtain, within each category of interest, the comparison between surveys. Additionally, in the 2016 survey, the variables of sex, age, educational level, drug use and type of community were analyzed as predictors of each attitude toward PWSUD, using the prevalence ratio model. Weighted data were used in all statistical analyses.

## 3. Results

As seen in Table 1, most participants in both surveys were in the age range of 30–65 years (53.9% in 2011 and 53.6% in 2016). In the 2011 sample, 7.8% had ever used illegal drugs or prescription drugs for non-medical purposes, as compared with 10.3% in 2016. Most of the respondents lived in urban communities (78.1% in 2011 and 77.7% in 2016). It is important to note that the sociodemographic distributions in each survey were similar, with similar proportions by sex and type of community, but the use of drugs in 2016 was greater than that in 2011.

Data analysis using the prevalence ratio (PR) model shows a significant drop from 2011 to 2016 in sympathetic attitudes: those regarding PWSUD as sick or in need of help (Table 2). The view that they were sick decreased in the total population (PR = 0.96, CI: 0.93–0.99, *p* = 0.005), among men (PR = 0.93, CI: 0.89–0.98, *p* = 0.002), among adults aged 30–65 (PR = 0.95, CI: 0.92–0.99, *p* = 0.005), among persons with drug use (PR = 0.89, CI: 0.79–0.99, *p* = 0.025) and non-users (PR = 0.96, CI: 0.93–1.00, *p* = 0.021), and among the urban population (PR = 0.93, CI: 0.90–0.97, *p* < 0.001); the belief that they were in need of help also dropped significantly in all groups. The findings for the negative attitudes, including the beliefs that PWSUD were weak, selfish or criminal, generally showed increases. The belief that they were weak showed significant increases, with differences between the population as a whole, women, teenagers and persons without drug use. The view that they were selfish increased in all groups except those who had used drugs. Finally, the attitude that PWSUD were criminals increased significantly in all groups except teenagers, those who had used drugs and rural dwellers.

Analysis of differences in perception according to the sociodemographic variables of interest in the 2016 survey (Table 3) shows that more women than men considered PWSUD to be sick (PR = 1.05, CI: 1.02–1.08, *p* = 0.004) or in need of help (PR = 1.09, CI: 1.06–1.12, *p* < 0.001), and they were less likely than men to perceive them as criminals (PR = 0.94, CI: 0.88–1.00, *p* = 0.045). More respondents over the age of 18 than those aged 12–17 believed they were sick (18–29: PR = 1.08, CI: 1.03–1.14, *p* = 0.002; 30–65: PR = 1.27, CI: 1.22–1.32, *p* < 0.001), weak (18–29: PR = 1.22, CI: 1.13–1.31, *p* < 0.001; 30–65: PR = 1.32, CI: 1.25–1.41, *p* < 0.001) or selfish (18–29: PR = 1.14, CI: 1.03–1.27, *p* = 0.010; 30–65: PR = 1.18, CI: 1.08–1.30, *p* < 0.001). More respondents aged 30–65 than teenagers perceived them to be criminal (PR = 1.10, CI: 1.03–1.19, *p* = 0.007), and more respondents who had never used drugs perceived them to be sick (PR = 1.15, CI: 1.08–1.23, *p* < 0.001), in need of help (PR = 1.11, CI: 1.05–1.17, *p* < 0.001) or criminal (PR = 1.24, CI: 1.11–1.39, *p* < 0.001) than those who had used drugs. More rural than urban residents viewed them as sick (PR = 0.96, CI: 0.92–0.99, *p* = 0.027) or selfish (PR = 0.87, CI: 0.78–0.98, *p* = 0.018). Finally, more people with a junior high school education (PR = 1.21, CI: 1.08–1.36, *p* = 0.001) and elementary education or less (PR = 1.32, CI: 1.16–1.50, *p* < 0.001) perceived them to be criminal than those with a higher educational level.

## 4. Discussion

One of the main contributions of this study is its analysis, for the first time in Mexico, of changes in attitudes toward PWSUD, with the use of probabilistic national household surveys that include both urban and rural communities. The findings indicate that negative social attitudes toward PWSUD increased over a period of five years, particularly among persons without drug use and those aged 30–65. At the same time, indicators of support for PWSUD (the attitude that they were sick or in need of help) decreased. The 2016 data also show more of these indicators of support among women than among men, and the view of them as criminals is more prevalent among those aged 30–65 than among teenagers. These differences are like those in studies in Spain and Germany, which showed more favorable or supportive attitudes among women, younger people and those with a higher educational level. They could be explained by a certain tendency in women to show greater empathy, as well as a more open attitude among young people toward PWSUD [18,29].

Our results for urban and rural populations show contradictions. The rural population shows a higher percentage both for the negative attitude that PWSUD are selfish and the positive attitude that they are sick. Although other studies have reported increases in favorable perceptions toward PWSUD, the prevailing tendency we find is an increase in negative perceptions [6,7,18,30,31]. Rodríguez et al. (2014) also found an increase over time in negative attitudes, such as thinking that PWSUD are criminals, which is similar to our finding.

These results could perhaps be explained by the current violence in Mexico surrounding drug trafficking and a consequent stigmatizing vision of persons who use drugs as criminals. In Mexico, growing perceptions of drug use as criminality pose a major challenge. A 2017 report by the Global Commission on Drug Policy argues that policies based on the criminalization of drug use respond to a vision of a moral rather than a public health problem, which could lead to delays in providing health care to dependent drug users who engage in high-risk behavior, generating more serious health problems. This attitude could also favor processes of exclusion and social marginalization. It is thus important to broaden efforts to provide public information from a public health perspective that can contribute to decision making [32].

In comparison with the data for 2011, results of the 2016 survey reveal a decrease in the general population’s perception of PWSUD as individuals with an illness in need of treatment. This attitude was seen both in respondents who had used drugs and those who had not, which could also have negative implications for the early attention and specialized treatment for such problems. The stigmatization of the people who use drugs can be a barrier to seeking treatment and also cause them to be denied treatment [13,14,15,16,21].

### 4.1. Limitations

The indicators used in this study are an initial approximation in the study of public attitudes toward addiction in representative samples of the Mexican population. It will be necessary in future research to include more indicators with more response options, as has been conducted in other studies, based on exploring social distance, the availability of help for substance users and other factors while maintaining a balance between sympathetic and negative beliefs and taking into account respondents’ emotional response toward those who use drugs, in order to provide psychoeducational resources for the general public. This study examined attitudes toward PWSUD but not toward those with dependence on alcohol; future research could also include attitudes toward alcohol use to establish possible differences by type of substance and their relationship with sociodemographic variables. Another possible limitation is the survey methodology of face-to-face interviews. It is possible in both surveys that participants’ responses were affected by assumptions of social desirability, given the nature of the topic.

### 4.2. Implications of This Study

It is crucial to encourage research along the lines of this study, not only in Mexico but throughout Latin America, with the goal of implementing a system for evaluating changes in attitudes toward PWSUD. Such a system would be useful for designing strategies for public information about the importance of mental health and the implications of stigmatization on the lives of individuals and families who are affected by substance abuse. Recent years have seen an increasing recognition of the importance of language in ending the stigmatization of people who use drugs [33]. Doing so is the social responsibility of families, communities, health care institutions and professionals, the media and, above all, those who make decisions and public policy regarding SUD. It is necessary to improve the care of substance users with an approach grounded in agency and recovery.

In addition, there is a need to develop interventions aimed at health personnel, school staff and the community, including the experiences of PWSUD, concerning the impact of stigma and discrimination directed at them and their families, which could contribute to raising awareness and informing society on these issues. Influence the development of public policies that are more inclusive and promote human rights and the inclusion of PWSUD. They include, of course, a gender perspective.

## 5. Conclusions

Derived from the results of our study, the main conclusions are:The groups with the most negative attitudes toward PWSUD were men, people 18 years of age or older, people who do not use drugs and people with lower educational levels.Women, young people, people with higher educational levels, people from urban areas and those who have used drugs presented more favorable attitudes toward PWSUD.There was an increase over time in negative attitudes towards PWSUD as selfish or criminal among all groups analyzed, except for those who use drugs.Among people who have used drugs, there was a decrease over time in the attitude that PWSUD are sick or in need of help, suggesting that they may not recognize that they have a health problem or that they could require professional attention.

We believe it is important to carry out additional studies in Mexico and other countries in Latin America to evaluate social perceptions toward PWSUD and expand our understanding of the cross-cultural implications of social stigma and SUD. This knowledge will allow us to design better preventive measures and provide psychoeducational interventions to the general population through the media, educational institutions and health care facilities. These actions can be especially important in addressing the stigmatization of women, members of the LGBTI community, homeless people and other groups who use drugs and are the objects of multiple discrimination. It is also necessary to work directly with people who use drugs to reduce their self-stigmatization, which can increase relapses and overdoses in PWSUD and hinder adherence to treatment [34,35,36,37].

## Figures and Tables

**Table 1 ijerph-19-08538-t001:** Sociodemographic structure of the two household surveys.

		2011			2016		
Variable		Men, *n* (%)	Women, *n* (%)	Total, *n* (%)	Men, *n* (%)	Women, *n* (%)	Total, *n* (%)
Age (years)	12–17	1389 (17.7)	1353 (16.1)	2742 (16.9)	4835 (17.6)	4728 (16.1)	9563 (16.8)
	18–29	2328 (29.6)	2428 (28.9)	4756 (29.3)	8330 (30.3)	8475 (28.8)	16,806 (29.5)
	30–65	4141 (52.7)	4609 (54.9)	8750 (53.9)	14,298 (52.1)	16,211 (55.1)	30,509 (53.6)
Drug use lifetime	Yes	1022 (13.0)	251 (3.0)	1272 (7.8)	4436 (16.2)	1423 (4.8)	5858 (10.3)
	No	6837 (87.0)	8140 (97.0)	14,977 (92.2)	23,027 (83.8)	27,991 (95.2)	51,019 (89.7)
Community	Rural	1776 (22.6)	1778 (21.2)	3555 (21.9)	6012 (21.9)	6671 (22.7)	12,683 (22.3)
	Urban	6082 (77.4)	6612 (78.8)	12,694 (78.1)	21,451 (78.1)	22,743 (77.3)	44,194 (77.7)

Survey 2011: *n* 16,249; survey 2016: *n* 56,877.

**Table 2 ijerph-19-08538-t002:** Attitudes among the general population toward persons with substance use disorder.

			Sick				Weak				Selfish				Needs Help				Criminal			
Variable	Year	*n*	%	PR	*p*-Value	95% CI	%	PR	*p*-Value	95% CI	%	PR	*p*-Value	95% CI	%	PR	*p*-Value	95% CI	%	PR	*p*-Value	95% CI
Total	2011	16,249	51.5	1.00			26.2	1.00			10.0	1.00			63.9	1.00			16.1	1.00		
	2016	56,877	49.1	**0.96**	0.005	0.93–0.99	28.0	**1.06**	0.035	1.01–1.12	13.7	**1.36**	<0.001	1.26–1.47	56.2	**0.88**	<0.001	0.85–0.91	19.5	**1.21**	<0.001	1.13–1.28
Men	2011	7859	51.2	1.00			27.4	1.00			10.1	1.00			61.8	1.00			16.7	1.00		
	2016	27,463	47.4	**0.93**	0.002	0.89–0.98	28.1	1.04	0.359	0.96–1.11	13.7	**1.39**	<0.001	1.26–1.52	53.4	**0.87**	<0.001	0.83–0.91	19.8	**1.21**	<0.001	1.11–1.31
Women	2011	8390	51.7	1.00			25.2	1.00			9.8	1.00			65.8	1.00			15.6	1.00		
	2016	29,414	50.7	0.98	0.247	0.94–1.02	28.0	**1.09**	0.019	1.02–1.16	13.7	**1.33**	<0.001	1.20–1.47	58.7	**0.89**	<0.001	0.86–0.92	19.2	**1.20**	<0.001	1.11–1.30
Age (years)																						
12–17	2011	2742	42.6	1.00			19.5	1.00			8.8	1.00			65.6	1.00			17.1	1.00		
	2016	9563	42.3	0.99	0.722	0.93–1.05	22.8	**1.15**	0.023	1.03–1.26	12.2	**1.32**	0.001	1.16–1.49	56.5	**0.86**	<0.001	0.82–0.90	18.9	1.09	0.147	0.97–1.21
18–29	2011	4756	47.4	1.00			26.8	1.00			10.2	1.00			65.3	1.00			14.9	1.00		
	2016	16,806	44.6	0.95	0.124	0.89–1.02	27.7	1.04	0.440	0.94–1.14	13.5	**1.35**	<0.001	1.18–1.51	56.0	**0.86**	<0.001	0.82–0.91	18.2	**1.25**	0.001	1.11–1.38
30–65	2011	8750	56.4	1.00			28.1	1.00			10.2	1.00			62.5	1.00			16.5	1.00		
	2016	30,509	53.7	**0.95**	0.005	0.92–0.99	29.9	1.06	0.096	0.99–1.12	14.3	**1.38**	<0.001	1.26–1.51	56.2	**0.90**	<0.001	0.86–0.93	20.4	**1.22**	<0.001	1.12–1.32
Drug use																						
Ever used	2011	1272	48.7	1.00			34.2	1.00			14.7	1.00			62.8	1.00			18.1	1.00		
	2016	5858	41.8	**0.89**	0.025	0.79–0.99	28.7	0.87	0.065	0.73–1.02	13.2	0.89	0.456	0.60–1.19	50.1	**0.80**	<0.001	0.72–0.88	15.9	0.86	0.200	0.64–1.09
Never used	2011	14,977	51.7	1.00			25.6	1.00			9.6	1.00			64.0	1.00			16.0	1.00		
	2016	51,019	50.0	**0.96**	0.021	0.93–1.00	28.0	**1.09**	0.004	1.03–1.15	13.8	**1.44**	<0.001	1.33–1.55	56.9	**0.89**	<0.001	0.86–0.91	19.9	**1.25**	<0.001	1.17–1.33
Community type																						
Rural	2011	3555	49.0	1.00			24.6	1.00			11.1	1.00			66.8	1.00			18.7	1.00		
	2016	12,683	51.8	1.05	0.151	0.98–1.11	27.3	1.09	0.146	0.97–1.22	15.5	**1.36**	0.002	1.17–1.56	57.2	**0.85**	<0.001	0.80–0.90	20.7	1.10	0.273	0.93–1.26
Urban	2011	12,694	52.2	1.00			26.7	1.00			9.6	1.00			63.0	1.00			15.4	1.00		
	2016	44,194	48.4	**0.93**	<0.001	0.90–0.97	28.3	1.06	0.104	0.99–1.12	13.2	**1.36**	<0.001	1.24–1.48	55.9	**0.89**	<0.001	0.86–0.92	19.2	**1.24**	<0.001	1.16–1.33

*PR*—prevalence ratio; CI—confidence interval. The sample for the 2011 household survey was *n* = 16,249; for 2016, it was *n* = 56,877. Values in bold are significant statistically.

**Table 3 ijerph-19-08538-t003:** Attitudes toward persons with substance use disorder in the 2016 household survey: prevalence ratio model.

			Sick				Weak				Selfish				Needs Help				Criminal			
Variable		*n*	%	PR	*p*-Value	95% CI	%	PR	*p*-Value	95% CI	%	PR	*p*-Value	95% CI	%	PR	*p*-Value	95% CI	%	PR	*p*-Value	95% CI
Sex	Men	27,463	47.4	1.00			28.1	1.00			13.7	1.00			53.4	1.00			19.8	1.00		
	Women	29,414	50.7	**1.05**	0.004	1.02–1.08	28.0	1.00	0.919	0.95–1.05	13.7	0.99	0.790	0.91–1.07	58.7	**1.09**	<0.001	1.06–1.12	19.2	**0.94**	0.045	0.88–1.00
Age	12–17	9563	42.3	1.00			22.8	1.00			12.2	1.00			56.5	1.00			18.9	1.00		
	18–29	16,806	44.6	**1.08**	0.002	1.03–1.14	27.7	**1.22**	<0.001	1.13–1.31	13.5	**1.14**	0.010	1.03–1.27	56.0	1.01	0.624	0.97–1.05	18.2	1.06	0.206	0.97–1.16
	30–65	30,509	53.7	**1.27**	<0.001	1.22–1.32	29.9	**1.32**	<0.001	1.25–1.41	14.3	**1.18**	<0.001	1.08–1.30	56.2	1.00	0.956	0.97–1.03	20.4	**1.10**	0.007	1.03–1.19
Educational level	College	8000	49.1	1.00			29.7	1.00			13.7	1.00			53.9	1.00			16.7	1.00		
	High school	14,552	45.9	0.96	0.173	0.91–1.02	27.6	0.96	0.395	0.88–1.05	12.9	0.96	0.551	0.83–1.11	56.2	1.04	0.20	0.98–1.09	17.8	1.08	0.240	0.95–1.23
	Junior school	20,033	48.4	1.00	0.904	0.94–1.05	28.0	1.00	0.939	0.92–1.08	13.4	0.99	0.898	0.87–1.13	56.7	1.04	0.122	0.99–1.09	20.0	**1.21**	0.001	1.08–1.36
	Elementary or less	12,930	53.5	1.04	0.123	0.99–1.10	28.1	0.96	0.423	0.88–1.05	15.1	1.08	0.309	0.93–1.25	57.0	1.04	0.122	0.99–1.10	22.2	**1.32**	<0.001	1.16–1.50
Drug use lifetime	Ever	5858	41.8	1.00			28.7	1.00			13.2	1.00			50.1	1.00			15.9	1.00		
	Never	51,019	50.0	**1.15**	<0.001	1.08–1.23	28.0	0.98	0.691	0.90–1.07	13.8	1.04	0.624	0.90–1.19	56.9	**1.11**	<0.001	1.05–1.17	19.9	**1.24**	<0.001	1.11–1.39
Community type	Rural	12,683	51.8	1.00			27.3	1.00			15.5	1.00			57.2	1.00			20.7	1.00		
	Urban	44,194	48.4	**0.96**	0.027	0.92–0.99	28.3	1.02	0.542	0.95–1.10	13.2	**0.87**	0.018	0.78–0.98	55.9	0.98	0.358	0.95–1.02	19.2	0.98	0.618	0.90–1.06

*PR*—prevalence ratio. Values in bold are significant statistically.

## Data Availability

Not applicable.

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
