# Peer review of "Changes in Attitudes toward People with Substance Use Disorder: A Comparative Study of the General Population in Mexico"

_ijerph, 2022, doi:10.3390/ijerph19148538_

Round 1

Reviewer 1 Report

Thank you for inviting me to review the manuscript "Changes in Attitudes toward People with Substance Use Disorder: A Comparative Study of the General Population in Mexico". This manuscript has an interesting topic and a solid methodology. I have some concerns over its organization and writing of the introduction, method and discussion part:

1. the introduction has several problems.  
(1) Please expand the first two paragraphs of the literature review to include sufficient studies related to SUD-related stigma. 
(2) The flow is not fluent. In lines 50-76, the authors seemed to review studies related to attitudes towards PWSUD. The paragraphs read lengthy and hard to follow. It is better to organize them into a paragraph or a section with top a top guiding sentence. A similar problem happened for lines 77-93, top sentences should be added to guide the literature review. 
(3) In the paragraph before line 108, a research gap should be constructed here that, the objective of the current study is to fulfill this gap. As a reader, I did not clearly spot the significance of the current study, which should be a contribution to fulfill a particular research gap constructed by the literature review.

2. the method part is not clear enough:
 (1) the first paragraph (lines 160-163) is introducing the key variable, the attitudes towards PWSUD, while its measurement is introduced in the last paragraph (lines 174-185). The two should be combined together. Also, line 166 was using "drug use" while SUD was used in the other part of the paper. Please keep the notions consistent. 
(2) the codings of all variables were not presented. 
(3) "2.4 Procedure" should be present together with "2.1" of design. "2.3 measurements" should be a subtitle of "2.2 questionnaire" (instead of instrument).

3. the discussion should be expanded to discuss its findings, including the similarity and differences between the current findings and the previous ones, especially those studies of SUD related stigma. 

Reviewer 2 Report

Bustos-Gamiño et al. conducted a study evaluating the general population's perception toward people with substance use disorders in Mexico. The topic is relevant for public health. However, the survey items considered in this study were very limited and did not allow a comprehensive evaluation of these perceptions. I have some suggestions to improve it:

Overall

The sentences are very long. I suggest making them shorter to facilitate their reading.

Abstract

I suggest defining the abbreviation of substance use disorders.

Introduction

Define stigma before mentioning the concept

The information regarding stigmatization might sound repetitive. Please, try to summarize it.

Page 2, line 63. Please, add the reference to the study mentioned  in this paragraph

Methods

This section could be better organized. Some relevant descriptions needed to understand the instruments or the surveys are found at the end of this section. I suggest adding each relevant description immediately after the first mention of the instruments/surveys.

Please, add the reference for the household survey methods

Please, add the reference for information mentioned in line 129

Is there any reference for the instrument used in the survey? Has this instrument previously been validated?

There were only two options (yes or no) for the items considered in this study. This does not give an option for people who prefer not to answer those specific items. Also, the number of items considered is very limited and does not allow the evaluation of other attitudes towards people with substance use disorders that could be relevant.

Results

In the tables, it would be helpful to highlight (perhaps in bold font) the significant results

Discussion

Please, elaborate on comparing the results from this study with previous ones, particularly those in Mexico.

Is there any proposed strategy to change people's negative perception toward people with substance use disorders?

Please, include in the study limitations that the number of items considered here is very limited and does not allow a comprehensive evaluation of other attitudes towards people with substance use disorders. Also, there were only two options (yes or no) for the items considered in this study, not allowing people to decline to answer these specific items.

Reviewer 3 Report

Overall this manuscript is very informative in demonstrating the changes in attitudes toward PWSUD in a large population scale.  Besides, I was wondering if the drug use history could be more specific to allow analysis of more conditions instead of the clean-cut 'ever used' vs 'never used' conditions. 

Author Response

Point 1. Overall, this manuscript is very informative in demonstrating the changes in attitudes toward PWSUD in a large population scale.  Besides, I was wondering if the drug use history could be more specific to allow analysis of more conditions instead of the clean-cut 'ever used' vs 'never used' conditions. 

Response: 

Thank you for your comment.

In Mexico, drug use is low. Lifetime prevalence was 7.8% in 2011 and 10.3% in 2016. For both studies, the prevalence of dependency in the last year was 0.6%. Finally, the prevalence in the last year was between 1.8% and 2.9%, which limits the quality of the results, significantly increasing the confidence intervals if we used a different consumption pattern with such small groups and having four additional predictor variables. 

Round 2

Reviewer 1 Report

Thank the authors for the revised version, and there were improvements. However, some problems were not solved:

  1.  the introduction is still in a poor structure. The first two paragraphs look good. However, from lines 45 - 108, the flow is still not clear. It was very informal to start a paragraph with a statement about a particular study. As I stated in the last round of the review, the top sentences should be used to guide the flow of the literature review. Simply listing studies are not cohesive.
  2. Similarly, as the top and conclusion sentences of each review section were absent, the research gap does not look significant. That is, just a sentence stating a research gap is not solid enough. The review logic, guided by top sentences and conclusions of each section, is not clear in constructing the research gap. 
  3. There are still problems in the method part. 
    1. In table 3, the line below "sex" should be "age", isn't it?
    2. Line 203, as each item is used separately, why did the author compute the cronbach's alpha?
    3. lines 205-206, it looks meaningless. I think the items should be employed only because there is an empirical reason.

Other problems

the writing was informal and looked careless, which is far from being a scientific publication. I sincerely suggest the authors to fully check the manuscript before submission. Just take some examples:

(1) Abbreviations. Authors should give the abbreviated version of terms in brackets, after the first occurrence of the full name, and then use abbreviations consistently in the following (e.g., lines 184, 190).

(2) Two colons appeared in the same sentence (e.g., line 180).

(3) Decimal places were not consistent in the whole paper. 

Reviewer 2 Report

No further comments

Author Response

We appreciate your comments.